# MDDM: Practical Message-Driven Generative Image Steganography Based on Diffusion Models

**Zihao Xu** [1]  **Dawei Xu** [1,2]  **Zihan Li** [2]  **Chuan Zhang** [2]

## Abstract

Generative image steganography (GIS) is an emerging technique that conceals secret messages in the generation of images. Compared to GAN-based or flow-based GIS schemes, diffusion model-based solutions can provide high-quality and more diverse images, thus receiving considerable attention recently. However, previous GIS schemes still face challenges in terms of extraction accuracy, controllability, and practicality. To address the above issues, this paper proposes a practical message-driven GIS framework based on diffusion models, called MDDM. Specifically, by utilizing the Cardan grille, we encode messages into Gaussian noise, which serves as the initial input for image generation, enabling users to generate diverse images via controllable prompts without additional training. During the information extraction process, receivers only need to use the pre-shared Cardan grille to perform diffusion inversion and recover the messages without requiring the image generation seeds or prompts. Experimental results demonstrate that MDDM offers notable advantages in terms of accuracy, controllability, practicality, and security. With flexible strategies, MDDM can achieve accuracy close to 100% under appropriate settings. Additionally, MDDM demonstrates certain robustness and potential for application in watermarking tasks.

## 1. Introduction

With the proliferation of the Internet and digital communications, information security issues have become increasingly prominent. As a crucial technique, steganography offers a secure and covert method of communication. With the increasing volume of image data on digital platforms, images have become a primary medium for covert data embedding (Bachrach & Shih, 2011; Subramanian et al., 2021; Mandal et al., 2022). Image steganography conceals information within images in a manner that prevents unauthorized access or detection. It is widely used in copyright protection (Altaay et al., 2012) and covert communication (Juneja, 2014).

However, achieving provably secure image steganography remains a major challenge. Traditional methods (van Schyndel et al., 1994; Liao et al., 2020; Yang et al., 2020; Su et al., 2021; Chan & Cheng, 2004) usually embed secret data through cover image modification. These methods often leave traces, making them susceptible to detection by steganalysis tools, and are therefore deemed empirically secure rather than provably secure (Hopper et al., 2002).

Provable security in image steganography requires that the distributions of the normal and stego images be indistinguishable (Weiming, 2023). Recently, the emergence of Artificial Intelligence Generated Content (AIGC) has provided a more diverse and flexible steganographic environment. Meanwhile, AI-generated data follows a controllable distribution. Leveraging this advantage, generative image steganography (GIS) has emerged and demonstrated strong resistance to typical steganographic attacks.

Compared with traditional methods, GIS schemes have better anti-detection performance against existing statistical feature-based steganalysis methods. GIS is mainly based on generative adversarial networks (GAN) (Yang et al., 2024b; Zhou et al., 2023a; Li et al., 2020; Su et al., 2024; You et al., 2022), flow-based models (such as Glow) (Zhou et al., 2023b; Wei et al., 2022; Xu et al., 2022) and the latest diffusion models (DM) (Hu et al., 2024; Peng et al., 2023; 2024; Yu et al., 2024; Jois et al., 2024). Although GIS has achieved notable progress, it still faces limitations in practical applications. For example, GAN-based methods require much training (Yang et al., 2024b), and are costly and difficult to control; flow-based methods require high computational resources during training and inference, especially in high-resolution image generation tasks (Zhou et al., 2023b). Diffusion model-based methods have made breakthroughs in generation quality and provide a better steganographic environment due to their widespread application in image generation. However, they still suffer

---

[1]Changchun University [2]Beijing Institute of Technology. Correspondence to: Chuan Zhang <chuanz@bit.edu.cn>.

*Proceedings of the 42ⁿᵈ International Conference on Machine Learning*, Vancouver, Canada. PMLR 267, 2025. Copyright 2025 by the author(s).

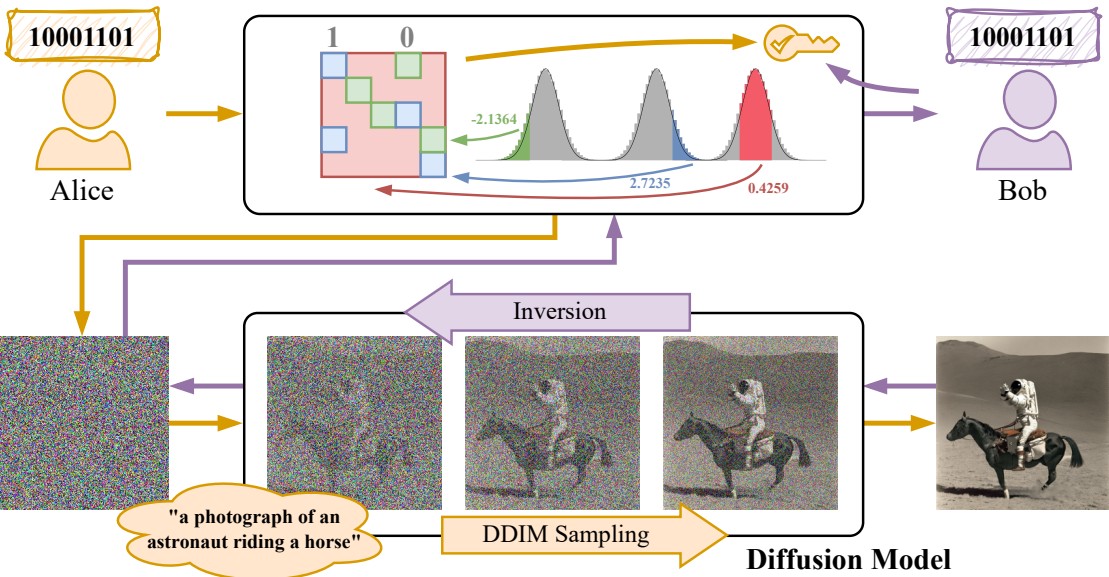

*Figure 1.* The overall framework of MDDM. The sender needs to share the Cardan grille with the receiver in advance to determine the location of the information in the initial noise. The Cardan grille is randomly generated. For the same message and the same Cardan grille, the sender can use MDDM to generate multiple images for selection.

from certain limitations. Specifically, diffusion-based generative image steganography falls into two groups: image-translation methods and methods that exploit the generative process. Image-translation methods keep the stego image visually close to the secret image, which limits output diversity (Yu et al., 2024; Yang et al., 2024a). Methods that exploit the generative process, aside from watermarking methods that usually hide only short messages, generally require reproducing the entire generation with the same seed and prompt (Peng et al., 2023; Jois et al., 2024; Peng et al., 2024); once these are shared, the sender's ability to produce arbitrary images is constrained and may require additional coordination.

To address these issues, we propose MDDM, a practical, message-driven generative image steganography framework (see Figure 1). Using a carefully designed encoding strategy, the sender generates a Cardan grille and maps a uniformly random binary message to Gaussian noise that follows the standard normal distribution. Then, the sender uses DDIM and a conditional prompt to generate a stego image starting from the initial noise. In this process, the sender may resample the noise while keeping the message and Cardan grille fixed, and vary the prompts to obtain diverse candidate images for selection. Because the procedure is identical to standard image generation, the stego images are indistinguishable from regular ones. The receiver inverts the stego image to recover the noise and, using the pre-shared Cardan grille, decodes the hidden message without the seed or prompt of image generation.

The main contributions of MDDM are as follows:

- **Message-driven GIS Framework**: We leverage the reversibility of DDIM to develop a message-driven image steganography framework based on diffusion models. By utilizing a Cardan grille, we encode a secret message into the initial noise for image generation. This enables MDDM to achieve imperceptible steganography.

- **Controllability and Practicality**: MDDM uses controllable conditional prompts and random seeds as input to the diffusion model, following the standard generation pipeline to produce high-quality, diverse stego images, while preserving secrecy and requiring no additional training. By adjusting the length of the secret message, MDDM can be flexibly applied to both information hiding and anti-counterfeiting watermarking.

- **Security and Robustness**: The images generated by MDDM are consistent with randomly generated images in distribution, which is considered to be provably secure. At the same time, Cardan grille in MDDM has good resistance to exhaustive attacks. In addition, a large number of experiments have shown that MDDM has certain robustness and can ensure a high extraction accuracy even under VAE compression and common image distortion conditions so that it can also be applied to watermarking tasks.

## 2. Related Work

The practice of image steganography has evolved considerably over time. With the advancement of digital technology, it has undergone significant evolution. The techniques have evolved from simple pixel modifications to the application of complex algorithms and advanced encryption methods, becoming increasingly sophisticated. In recent years, research has focused on improving concealment, enhancing security, and increasing resistance to attacks. Additionally, the application of artificial intelligence has further promoted innovation in image steganography.

### 2.1. Cover Image Based Steganography

From early research to the present day, most studies on image steganography have focused on embedding information directly into images. The oldest and most classic embedded steganography scheme is LSB (van Schyndel et al., 1994; Chan & Cheng, 2004), which embeds information into the least significant bit of image pixels. Furthermore, researchers have proposed manually designed methods (Liao et al., 2020; Su et al., 2021) and neural network-based techniques (Yang et al., 2020) as adaptive methods to reduce image distortion caused by embedding. There are also frequency domain-based steganography methods, including DctDwt (Al-Haj, 2007) and DctDwtSvd (Navas et al., 2008). In recent years, deep learning based steganography (Tancik et al., 2020; Guan et al., 2022; Lu et al., 2021; Xu et al., 2022; Jing et al., 2021; Zhu et al., 2018; Fadhil et al., 2023) has emerged. The first end-to-end trainable framework for data hiding, HIDDEN (Zhu et al., 2018), achieves embedding and extraction through autoencoders. Recent research on embedded steganography has largely focused on invertible neural networks (INNs) (Guan et al., 2022; Lu et al., 2021; Xu et al., 2022; Jing et al., 2021). For example, HiNet (Jing et al., 2021) is the first attempt to utilize invertible neural networks for image hiding tasks, where information hiding and extraction can be accomplished through a single network. However, the aforementioned embedded steganography schemes often leave modification traces on the image, which can still be detected (Fu et al., 2024).

### 2.2. Generative Image Steganography

Generative image steganography has gained traction and is generally considered more secure than traditional embedding-based methods (Yang et al., 2019). Generative image steganography, leveraging generative models, is commonly divided into three main categories: flow-based (Zhou et al., 2023b; Wei et al., 2022; Xu et al., 2022), GAN-based (Yang et al., 2024b; Zhou et al., 2023a; Li et al., 2020; Su et al., 2024; You et al., 2022), and diffusion-based methods. Flow-based methods primarily exploit the reversible properties of such models, with SR2IT (Zhou et al., 2023b) being

a representative study. Compared to flow-based methods, GAN-based research has been explored more extensively. For example, PARIS (Yang et al., 2024b) developed a provably secure method to image steganography and attached a noise module to the generator to improve robustness. Compared to GAN-based methods, diffusion models (Hu et al., 2024; Peng et al., 2023; 2024; Yu et al., 2024; Jois et al., 2024) have recently developed generative image steganography methods due to their advantages in generation quality and diversity. Currently, most research can be divided into two categories: image translation methods and methods that leverage the generative process. However, these methods still face challenges in computational efficiency, practical deployment, and steganographic capability.

### 2.3. Cardan Grille

The Cardan grille is a classical steganographic technique, typically used as a shared key between sender and receiver for hiding and extracting information, and has considerable security (Utepbergenov et al., 2013). Recently, some studies have also tried to apply it to generative image steganography. Most of these studies are based on the Cardan grille and use GAN to perform image restoration to generate stego images (Liu et al., 2018; Zhang et al., 2019c; Wang et al., 2021). Although these methods have demonstrated promising results, particularly in terms of security, there are still problems such as uncontrollable generation effects, low stego capacity, and poor versatility.

### 2.4. Diffusion Models

The rapid advancement of diffusion models has led to a surge in AI-generated images across the internet. Early diffusion models were primarily based on pixel-space diffusion like DDPM (Ho et al., 2020). However, in recent years, models based on latent diffusion, such as Stable Diffusion (Rombach et al., 2022), have become the dominant methods. DDIM (Song et al., 2020a), a widely used sampling method, has been applied to both unconditional and conditional diffusion models. Notably, DDIM inversion techniques have recently been introduced, enabling fine-grained editing of AI-generated images (Wallace et al., 2023; Zhang et al., 2025; Wang et al., 2024; Mokady et al., 2023).

## 3. Method

We regard the message hiding process Gen as generating a natural image from secret noise and the extraction process Ext as recovering the noise from a generated image:

$$\begin{aligned} \texttt{Gen}\left(sk, m\right) &= I_{stego}, \\ \texttt{Ext}\left(sk, I_{stego}\right) &= m, \end{aligned} \tag{1}$$

where $sk$ is the secret key, $m$ is the secret message, and $I_{stego}$ is the generated image.

In the following paragraphs, we provide a detailed description of our implementation of the message-driven diffusion model steganography (MDDM).

### 3.1. The Basics of MDDM

In our work, we exploit the properties of DDIM (Song et al., 2020a) to construct the information hiding and extraction process. DDIM is an improved diffusion model that, unlike DDPM (Ho et al., 2020), defines a non-Markov forward process. DDIM can improve the speed and quality of image generation and is always used as a sampling method. We exploit the following properties of DDIM to design our MDDM.

**Deterministic Sampling.** For steganography, we are particularly interested in the backward process of DDIM, which is a key distinction between DDIM and DDPM. We denote the number of diffusion steps as $T$, the initial noise as $x_T$, the indices of intermediate steps decrease progressively, and the final result of diffusion as $x_0$. The sampling formula is as follows:

$$x_{t-1} = \sqrt{\alpha_{t-1}} \left( \frac{\boldsymbol{x}_t - \sqrt{1-\alpha_t}\epsilon_\theta^{(t)}(\boldsymbol{x}_t)}{\sqrt{\alpha_t}} \right) \\ + \sqrt{1-\alpha_{t-1}-\sigma_t^2} \cdot \epsilon_\theta^{(t)}(\boldsymbol{x}_t) \\ + \sigma_t\epsilon_t. \quad (2)$$

When $\sigma_t = 0$, randomness disappears from the formula and this process becomes deterministic. The initial noise $x_T$ completely determines the final result $x_0$, which is a one-to-one relationship. In our work, we adopt deterministic DDIM sampling across various diffusion models.

**The Diffusion Inversion.** To "reverse" the diffusion sampling process is known as inversion. In practice, inversion is primarily employed for image editing, image interpolation, and image restoration (Wang et al., 2024). Inversion aims to recover the initial noise corresponding to an image, and under ideal conditions, if this initial noise is used as the starting point of the sampling process, the original image can be reproduced. A representative method is DDIM inversion, which recovers the initial noise by deterministically adding noise to the image.

However, prior studies have shown that DDIM inversion typically performs well in the unconditional case but is unreliable in the case of text-guided generation (Mokady et al., 2023; Wallace et al., 2023; Wang et al., 2024). Although our primary objective is steganography, in editing workflows involving conditional image generation we adopt exact inversion methods, such as EDICT (Wallace et al., 2023), which achieves exact diffusion inversion via coupled transformations. Notably, any inversion method is acceptable, as the choice of inversion method is not central to our study.

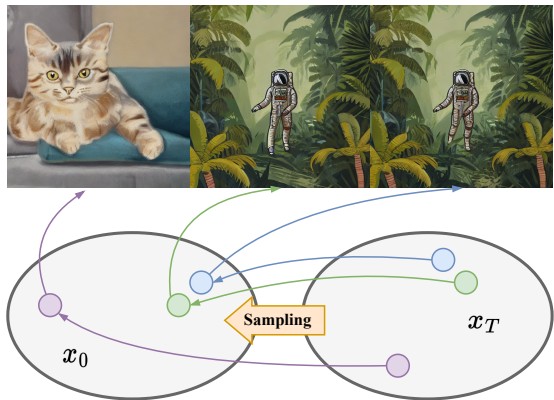

*Figure 2.* If the initial noise inputs are similar, then the images generated through deterministic sampling are also similar.

**Correlation between Generated Image and Initial Noise.** DDIM sampling is a deterministic process, which means that under the same conditions, two inputs with slight differences will follow almost identical sampling trajectories, as shown in Figure 2. Let $x_T^{(1)}$ and $x_T^{(2)}$ denote two Gaussian noise inputs with a small difference:

$$\|x_T^{(1)} - x_T^{(2)}\|_2 \approx \delta, \quad (3)$$

where $\delta$ represents a small difference.

Since the space of the input is continuous (Song et al., 2020b), $x_T^{(1)}$ and $x_T^{(2)}$ will undergo similar sampling processes and thus yield similar output images $x_0^{(1)}$ and $x_0^{(2)}$. As illustrated in Figure 3, when the noise follows the standard normal distribution, slight perturbations applied to select regions of the initial noise do not significantly alter the content of the generated image. Meanwhile, by employing the inversion method mentioned previously, the bias in the inversion process is effectively eliminated. Consequently, similarity in initial noise inputs correlates with similarity in the generated images.

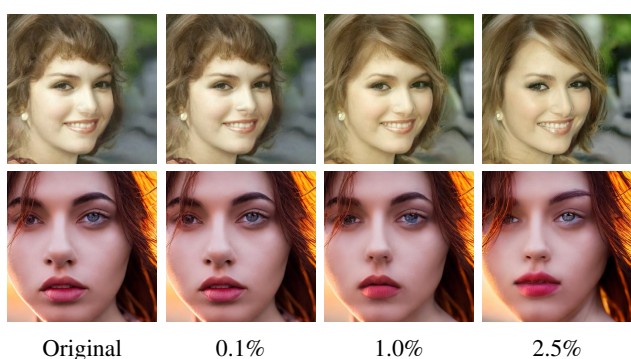

| Original | 0.1% | 1.0% | 2.5% |

*Figure 3.* Images generated using DDIM sampling on DDPM (top row) and Stable Diffusion (bottom row), with different perturbation ratios applied to the initial noise inputs.

**The Information Loss Is Acceptable.** The encoding and decoding process of an image inevitably leads to information loss, but our research shows that this loss is within an acceptable range. We study the loss of conditional image generation based on the Stable Diffusion model (SD) and unconditional image generation based on the Latent Diffusion Models (LDMs)[1] and DDPM. The original noise is called $x_T$, and the inverted noise is called $x_T'$. In order to evaluate the information loss, we flatten the high-dimensional space into a one-dimensional representation and calculate the absolute difference $|x_T'[i] - x_T[i]|$ element by element, as shown in Figure 4. The results show that, whether based on the latent diffusion or the pixel-space diffusion, the absolute difference between the original noise and the inverted noise is smaller than that of random noise and most of them remain within 1.0. It is worth noting that the pixel-space diffusion model may be unstable, and the range of absolute differences is slightly larger than that of the latent diffusion models. Since image generation based on latent diffusion models is more widely used in practical applications, our method mainly focuses on latent diffusion models.

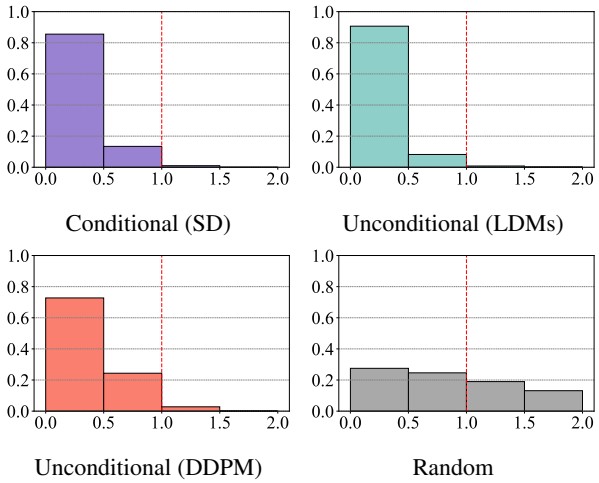

*Figure 4.* Distribution of the element-wise absolute difference between the initial and the inverted noise under different conditions (x-axis: absolute difference; y-axis: proportion).

### 3.2. Framework of MDDM

MDDM requires no additional training and instead relies on multiple pre-trained modules. Specifically, it leverages either latent diffusion models (e.g., Stable Diffusion) or pixel-space diffusion models (e.g., DDPM), together with a DDIM scheduler and inversion methods (such as DDIM inversion and EDICT), all of which are readily available.

In MDDM, the sender and receiver first share a randomly generated Cardan grille, equivalent to $sk$ in Equation (1), which specifies the location of the hidden message but does

---

[1]https://github.com/CompVis/latent-diffusion

not contain the message itself, as shown in Figure 5. Using our encoding strategy, the binary message is mapped to noise that follows the standard normal distribution. Then, the pre-trained diffusion model can generate images using this noise. Since the entire process mirrors general image generation, the stego image remains indistinguishable from a normally generated image. The receiver uses the received stego image to perform diffusion inversion, reconstructing the noise and recovering the hidden binary data through the pre-shared Cardan grille without requiring knowledge of the image generation seed or prompt. The benefit of MDDM is that the sender can generate multiple images without changing the secret information and Cardan grille and select the images with high information extraction accuracy and good quality for transmission.

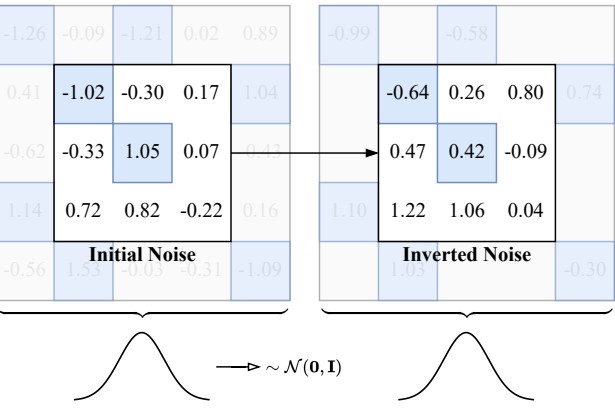

*Figure 5.* Visualization of the Cardan grille (blue grid area), which serves as a symmetric key and effectively preserves the distribution of the noise.

**Message Hiding.** In the hiding stage, considering that information loss may reduce the accuracy of steganographic content extraction, we first determine the area suitable for steganography. As shown in Section 3.1, for both latent diffusion models and pixel-space diffusion models, the element-wise absolute difference between the initial noise and the noise recovered by diffusion inversion typically falls within $[0, 1]$. Therefore, in order to ensure that potential errors do not affect the extraction accuracy, we define the highly robust area of steganography as data with the standard normal distribution in the range of $(-\infty, -1)$ and $(1, +\infty)$. Here, we introduce the standard normal distribution cumulative distribution function:

$$\Phi(x) = \int_{-\infty}^{x} \frac{1}{\sqrt{2\pi}} e^{-\frac{t^2}{2}} \, dt. \tag{4}$$

For a noise of size $1 \times c \times h \times w$ (Batch $\times$ Channel $\times$ Height $\times$ Width), the amount of data in the highly robust area $L_{max}$ can be calculated by the following formula:

$$L_{max} = 2 \times (1 - \Phi(1)) \times 1 \times c \times h \times w. \qquad (5)$$

For example, for noise of size $1 \times 4 \times 64 \times 64$, there are 5199 elements in the highly robust area. When we determine that the length $l$ of the secret message $m$ to be transmitted is less than $L_{max}$, we can calculate the truncation threshold $k$ by the inverse function of $\Phi(x)$ as follows:

$$k = \Phi^{-1}(1 - \frac{l}{2 \times 1 \times c \times h \times w}). \qquad (6)$$

Next, the Cardan grille $CG$ is a list of length $l$ with unique elements, generated by randomly sampling from the integer set $\{1, 2, \ldots, 1 \times c \times h \times w\}$. Let $\{\zeta_r\}_{r=1}^{1 \times c \times h \times w}$ denote the elements of this integer set, then:

$$CG = \texttt{RandomSample}\big(\{\zeta_r\}_{r=1}^{1 \times c \times h \times w}, l\big), \qquad (7)$$

where $\texttt{RandomSample}(\cdot, l)$ denotes selecting $l$ distinct elements at random.

It is worth noting that the secret message $m$ is a uniformly distributed binary string formed by encryption or other means. Since this process can be implemented in many ways and is not the focus of this article, this article does not describe how to generate a uniformly distributed binary string in detail.

At this point, the preparatory steps are complete. We next generate the initial noise by performing truncated sampling from the standard normal distribution at the previously determined Cardan grille positions $CG$. In each iteration, we draw a fresh batch of samples from $\mathcal{N}(\mathbf{0}, \mathbf{I})$ and partition them into three pools according to the truncation threshold $k$. For each bit of the secret message $m$, we then randomly sample from the $(-\infty, -k]$ pool if the bit is 0, or from the $(k, +\infty)$ pool if the bit is 1, and insert the sampled values into the one-dimensional noise vector at the indices specified by $CG$. After filling all message-bearing positions, the remaining entries are populated by sampling from the truncated interval $(-k, k]$ for each new draw and placing them into the unused slots of the noise vector. Finally, we reshape this one-dimensional vector to the diffusion model's required noise size $1 \times c \times h \times w$. In Section 4, we demonstrate that the constructed noise is statistically indistinguishable from samples drawn from the standard normal distribution, thereby offering a provable security guarantee for our method. Algorithm 1 details this procedure, and Appendix A presents an optimized variant designed to accelerate noise generation for high-dimensional inputs.

After applying the above encoding rules to map the secret message $m$ to the initial noise $x_T$, the sender directly uses $x_T$ for DDIM sampling, where the prompt can be arbitrary,

---

**Algorithm 1** Generate initial noise

---

**Input:** The noise size $(1, c, h, w)$, secret message $m$ with length $l$, Cardan grille $CG$, truncation threshold $k$
**Output:** Initial noise $x_T$
**Function** $F_1(n \in \{0, 1\})$
$list \sim \mathcal{N}(\mathbf{0}, \mathbf{I})$
**if** n = 0 **then**
    $y = \texttt{RandomChoice}\{u \mid u < -k, u \in list\}$
**else**
    $y = \texttt{RandomChoice}\{u \mid u > k, u \in list\}$
**end if**
**Return** $y$
**Function** $F_2()$
$list \sim \mathcal{N}(\mathbf{0}, \mathbf{I})$
$y = \texttt{RandomChoice}\{u \mid -k \leq u \leq k, u \in list\}$
**Return** $y$
Init noise $x_T$
**for** $i < l$ **do**
    $x_T[CG[i]] = F_1(m[i])$
**end for**
**for** $j < 1 \times c \times h \times w$ **do**
    **if** $j \notin CG$ **then**
        $x_T[j] = F_2()$
    **end if**
**end for**

---

resulting in the stego image $I_{stego}$. Taking the Stable Diffusion model as an example, the process can be formulated as:

$$I_{stego} = \mathcal{D}\left(\texttt{DDIMSampling}\left(S, Text, x_T\right)\right), \qquad (8)$$

where $\mathcal{D}$ denotes the VAE decoder, $\texttt{DDIMSampling}$ denotes the DDIM sampling process, $S$ denotes the number of DDIM sampling steps, $Text$ denotes the input prompt, and $x_T \sim \mathcal{N}(\mathbf{0}, \mathbf{I})$ denotes the initial noise.

**Message Extraction.** The receiver obtains the stego image and applies the inversion to quickly recover the inverted noise, without reconstructing the image. Using only the pre-shared Cardan grille, and without the need for any seeds or prompts, the receiver can sequentially extract the corresponding elements from the inverted noise and reconstruct the binary message based on whether each extracted value is greater than or less than zero:

$$m'[i] = \begin{cases} 0, & \text{if} \quad x'_T[CG[i]] \leq 0, \\ \\ 1, & \text{if} \quad x'_T[CG[i]] > 0. \end{cases} \qquad (9)$$

Ideally, the $m'$ obtained by the receiver's inversion should be consistent with the secret message $m$ hidden by the sender.

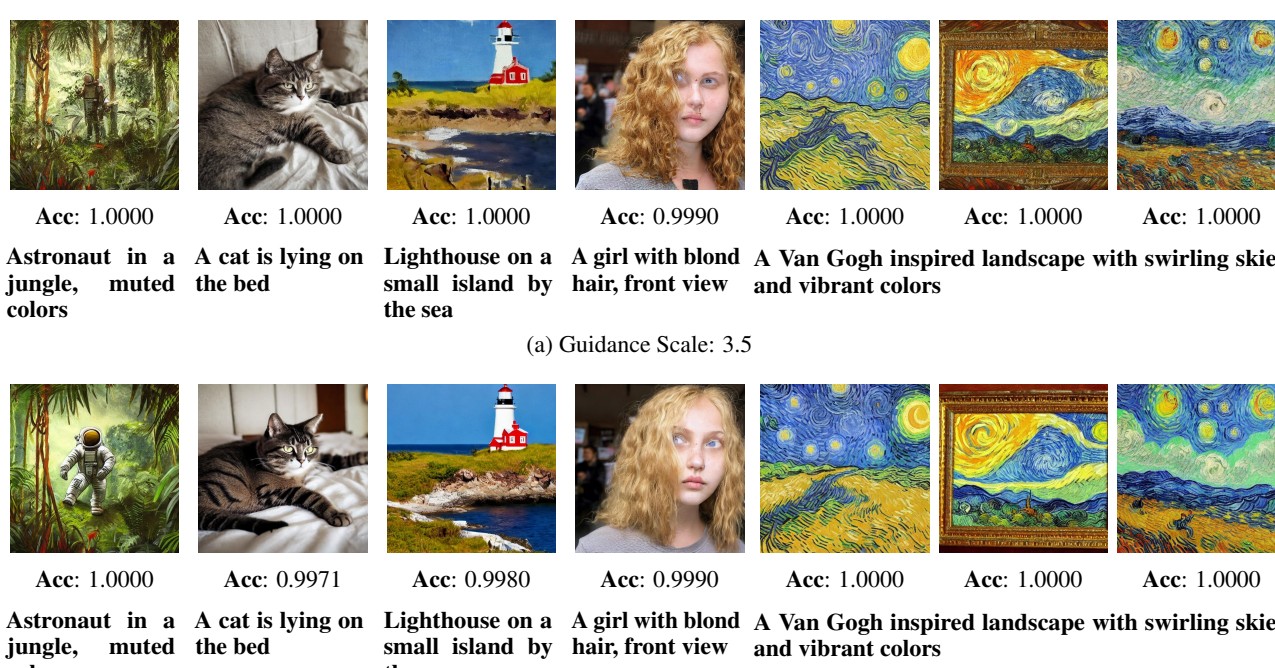

Acc: 1.0000    Acc: 1.0000    Acc: 1.0000    Acc: 0.9990    Acc: 1.0000    Acc: 1.0000    Acc: 1.0000

**Astronaut in a jungle, muted colors**   **A cat is lying on the bed**   **Lighthouse on a small island by the sea**   **A girl with blond hair, front view**   **A Van Gogh inspired landscape with swirling skies and vibrant colors**

(a) Guidance Scale: 3.5

Acc: 1.0000    Acc: 0.9971    Acc: 0.9980    Acc: 0.9990    Acc: 1.0000    Acc: 1.0000    Acc: 1.0000

**Astronaut in a jungle, muted colors**   **A cat is lying on the bed**   **Lighthouse on a small island by the sea**   **A girl with blond hair, front view**   **A Van Gogh inspired landscape with swirling skies and vibrant colors**

(b) Guidance Scale: 7.5

*Figure 6.* The performance of MDDM under different guidance scales. The default guidance scale setting of 7.5 can generally achieve better visual effects. The upper and lower images in each column are generated using the same seed.

## 4. Security Analysis

We first examine how MDDM steganography affects the distribution of generated images to demonstrate its provable resistance to steganalysis. To formally describe the sampling process, let $x_T \in \mathbb{R}^{1 \times c \times h \times w}$ denote the initial noise, and let $CG$ from Equation (7) denote the set of indices corresponding to the Cardan grille positions, whose size equals the length $l$ of the binary string $m$. For each index $i \in CG$, we embed one bit $m[i] \in \{0, 1\}$ by sampling the corresponding value $x_T^{(i)}$ independently from a truncated standard normal distribution. Specifically, we define the conditional distribution for $x_T^{(i)}$ as follows:

$$x_T^{(i)} \sim \begin{cases} \mathcal{N}(\mathbf{0}, \mathbf{I}) \text{ truncated to } (k, \infty), & \text{if } m[i] = 1, \\ \mathcal{N}(\mathbf{0}, \mathbf{I}) \text{ truncated to } (-\infty, -k], & \text{if } m[i] = 0. \end{cases} \tag{10}$$

The value of threshold $k$ is chosen according to the ratio of the payload length to the total embedding capacity, such that larger payloads require smaller $k$ to maintain coverage, while smaller payloads allow for more extreme truncation and thus potentially stronger security. For all remaining positions $j \notin CG$, we sample the values $x_T^{(j)}$ independently from the central region of the same distribution:

$$x_T^{(j)} \sim \mathcal{N}(\mathbf{0}, \mathbf{I}) \text{ truncated to } (-k, k]. \tag{11}$$

Since all samples are drawn independently, and the truncation preserves the symmetry and local density properties of the original distribution, the resulting noise vector $x_T$ maintains the overall statistical characteristics of the standard normal distribution. Moreover, because the Cardan grille positions are chosen uniformly at random and the embedded bits are uniformly distributed, the sampling strategy does not introduce any detectable structural bias. As such, the steganographic modification remains statistically indistinguishable from standard noise under conventional detection methods. According to Appendix B, MDDM is provably secure.

Next, we need to prove that the Cardan grille used by MDDM possesses privacy security. According to Appendix C, even when reusing the Cardan grille for steganography across multiple images, a third party cannot obtain the ordering privacy information of the Cardan grille, thus unable to threaten the privacy security of the hidden information.

## 5. Experiments

Our experiments are conducted based on publicly available pre-trained diffusion models. For unconditional image generation, we use DDIM for sampling and apply DDIM inversion as the inversion method. For conditional image generation, given that EDICT exhibits suboptimal sampling

*Table 1.* Comparison of the robustness of MDDM and baseline methods in terms of extraction accuracy for $512 \times 512$ images. The evaluation includes results for the "PNG" image (i.e., no attack), along with various attacks such as "JPEG" (image compression with an image quality of 95%), "Resize" (image scaling by a factor of 0.5), "Gaussian Blur (GB)" (with a radius of 1.0), "Drop" (randomly adding a white square of 0.5% area), "Brightness" (with a factor of 0.15), and "Rotation" (an angle of 0.3 degrees).

| Method | Version | Capacity (Bits) | PNG | JPEG (95) | Resize (0.5) | GB (1.0) | Drop (0.005) | Brightness (0.15) | Rotation (0.3) |
|---|---|---|---|---|---|---|---|---|---|
| DwtDct (Al-Haj, 2007) | - | 32 | 0.83 | 0.65 | 0.63 | 0.68 | 0.83 | 0.83 | 0.72 |
| DwtDctSvd (Navas et al., 2008) | - | 32 | 1.00 | 1.00 | 1.00 | 1.00 | 1.00 | 0.78 | 1.00 |
| RivaGAN (Zhang et al., 2019a) | - | 32 | 0.99 | 0.99 | 0.99 | 0.99 | 0.99 | 0.99 | 0.99 |
| LaWa (Rezaei et al., 2024) | SD-V1.4 | 48 | 1.00 | 1.00 | 1.00 | 0.99 | 1.00 | 1.00 | 1.00 |
| MDDM-LC (Ours) | SD-V1.4 | 48 | 1.00 | 1.00 | 0.99 | 1.00 | 1.00 | 1.00 | 0.99 |
| GS (Yang et al., 2024c) | SD-V2.1 | 256 | 1.00 | 1.00 | 1.00 | 1.00 | 1.00 | 1.00 | 1.00 |
| MDDM-LC (Ours) | SD-V2.1 | 256 | 1.00 | 1.00 | 0.99 | 1.00 | 1.00 | 1.00 | 0.99 |

*Table 2.* Comparison of steganographic capability among three MDDM variants under varying truncation thresholds.

| Type | Truncation Threshold | Bits | Accuracy | | |
|---|---|---|---|---|---|
| | | | Face | Bedroom | Cat |
| MDDM-PU3 | 2 | 8946 | 0.9847 | 0.9641 | 0.9816 |
| $256 \times 256$ | 1 | 62386 | 0.9221 | 0.8708 | 0.9581 |
| MDDM-LU | 2 | 559 | 0.9977 | 0.9951 | - |
| $256 \times 256$ | 1 | 3899 | 0.9890 | 0.9877 | - |
| MDDM-LC | 2 | 745 | 0.9987 | 0.9973 | 0.9988 |
| $512 \times 512$ | 1 | 5199 | 0.9946 | 0.9920 | 0.9913 |

the image generation effect and message accuracy achieve a high level. Based on this, in subsequent experiments based on Stable Diffusion, we use the guidance scale of 7.5 by default to conditionally generate stego images. For conditional image generation, we also evaluate the impact of prompts, as detailed in Appendix D.1. Figure 10 provides additional examples of generated images, demonstrating that our results are of high quality and diversity. In addition, we also evaluate the FID (Heusel et al., 2017) on the generated images, as detailed in Appendix D.2.

## 5.2. Steganographic Capability

We first evaluate the relationship between steganographic capacity and extraction accuracy (Acc) across three MDDM variants under varying truncation thresholds, as summarized in Table 2. Here, accuracy represents the correctness of the extracted message, while capacity reflects the amount of information that can be embedded. The specific settings of experiments are detailed in Appendix D.3. Experimental results indicate that as the length of the secret message increases, extraction accuracy declines but remains at a reasonable level. In addition, compared with the pixel-space diffusion models, the latent diffusion models achieve higher extraction accuracy. This further confirms our findings in Section 3.1. Therefore, our method is more suitable for image generation based on latent diffusion models. Moreover, we find that although variations in CPU and GPU configurations may yield differences in the initially generated noise, these variations have no significant impact on the overall performance of MDDM.

In Appendix D.4, we compare MDDM with the latest diffusion model-based generative image steganography methods under identical conditions. All compared methods involve hiding binary messages. The above experiments demonstrate the effectiveness of our proposed method.

quality (Wang et al., 2024) and, under the standard generative process, generations similar to DDIM (Wallace et al., 2023) while requiring more time, we therefore continue to use DDIM for sampling and adopt EDICT as the inversion method. The number of steps for both sampling and inversion is set to 50. All experiments are conducted on NVIDIA RTX 3090 GPUs.

Based on the type of generation, MDDM can be categorized into three variants: MDDM-PU for unconditional image generation with pixel-space diffusion models, MDDM-LU for unconditional image generation with latent diffusion models, and MDDM-LC for conditional image generation with latent diffusion models. Specifically, MDDM-PU leverages the publicly available pre-trained DDPM model from Hugging Face[2], MDDM-LU utilizes pre-trained LDMs, and MDDM-LC typically adopts Stable Diffusion v1.5.

## 5.1. Image Generation

Figure 6 compares conditional image generation results under different guidance scales. These images hide 1024 bits of information at a size of $512 \times 512$. The visualization results indicate that at the default guidance scale of 7.5,

---

[2]https://huggingface.co/

## 5.3. Robustness

In Section 3.1, we demonstrate a correlation between the similarity of the initial noise and the similarity of the final generated image. Consequently, when the final generated image remains largely unchanged, MDDM is not significantly affected, particularly by common image distortions such as JPEG compression. As shown in Table 1, MDDM resists various image-based attacks and offers more flexible information capacity. Note that since DwtDct, DwtDctSvd, and RivaGAN are not generative image watermarking methods, we use Stable Diffusion to generate cover images. To compare other generative image watermarking methods based on latent diffusion, we use prompts from Stable-Diffusion-Prompts [3]. Results indicate that MDDM can, to some extent, serve as a watermarking method.

## 5.4. Controllability and Practicality

MDDM enables random image generation while keeping the secret message and Cardan grille unchanged, allowing the sender to continuously generate diverse images for selection, as shown in Figure 10. Table 5 in Appendix D.5 presents a comparison of our method with other methods in terms of diversity, where diversity refers to the number of images that can be generated after a single communication and the sharing of a secret key or seed. Results show that existing regeneration-based methods (Peng et al., 2023; Jois et al., 2024; Peng et al., 2024) require changing the noise seed if the generated image is unsatisfactory, which may lead to increased communication overhead and potential risks. In contrast, our method synchronizes the encryption key for the message and the random seed for the Cardan grille during initial setup. After that, subsequent exchanges no longer require direct point-to-point communication. For example, the sender can use the initial seed to generate the Cardan grille and then post the stego images to their social media page. The receiver can then download these images and extract the information using the initial seed. For subsequent transmissions, the seed can be changed sequentially, and the receiver can do the same; thus, communication can proceed even asynchronously. In summary, with MDDM, each image sent by the sender is arbitrarily controllable, and multiple images can be generated for selection, which avoids arousing suspicion from potential attackers and ensures the accuracy of message extraction. This further demonstrates the safety and practicality of MDDM.

## 5.5. Resistance to Steganalysis

We test MDDM on ZhuNet (Zhang et al., 2019b), SiaStegNet (You et al., 2020), XuNet (Xu et al., 2016), YeNet

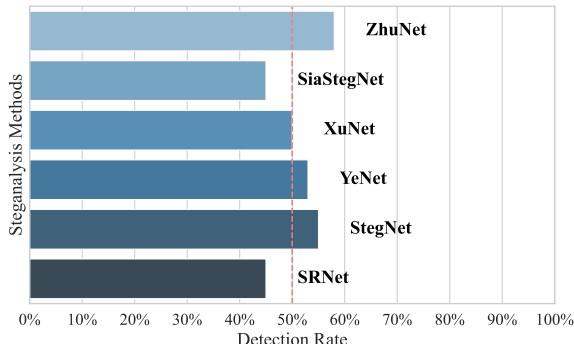

*Figure 7.* Results of detection using multiple advanced steganalysis methods.

(Ye et al., 2017), StegNet (Deng et al., 2019) and SRNet (Boroumand et al., 2018), as shown in Figure 7. The results are generally around 50%, indicating that the distribution cannot be distinguished, which is consistent with the theoretical analysis.

## 6. Conclusion

We propose MDDM, a practical message-driven generative image steganography method based on diffusion models. The process of generating stego images using MDDM is identical to the general image diffusion generation process, and MDDM performs well in terms of controllability, security, image quality, robustness, and extraction accuracy. In the future, we will introduce error-correcting codes and explore the performance of MDDM in other diffusion models.

## Acknowledgements

This work is partially supported by the 111 Project under Grant B21044, the Sichuan Science and Technology Program under Grants 2021ZDZX0011, the National Natural Science Foundation of China under Grant 62376175 and 62472032, and the Young Elite Scientists Sponsorship Program by CAST (Grant No. 2023QNRC001).

## Impact Statement

Generative image steganography (GIS) has recently emerged as a prominent research direction within the broader field of steganography. The research presented in this paper not only advances the development of GIS but also demonstrates the potential for real-world applications. For instance, the proposed MDDM is training-free, which may facilitate the rapid adoption of GIS technologies. Overall, our work is anticipated to make an impact in domains such as information security and digital copyright protection.

---

[3]https://huggingface.co/datasets/Gustavosta/Stable-Diffusion-Prompts

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

## A. Improved Algorithm for Initial Noise Generation

Analogous to the original algorithm, the enhanced method still uses each uniformly drawn binary bit (0 or 1) to decide whether its associated sample should lie in the left tail ($x \leq -k$) or the right tail ($x > k$) of the standard normal distribution. The key innovation is the adoption of vectorized rejection sampling for the central region: a large batch of standard normal variates is generated in a single operation, values falling within the interval $(-k, k]$ are identified via Boolean masking, and those accepted samples are filled sequentially into all non-tail positions. By avoiding element-wise loops for random-number generation, this method greatly improves overall execution efficiency. For the tail regions, the algorithm similarly generates multiple candidate variates in batches for each tail position, filters the subset satisfying the prescribed left or right tail condition, and randomly selects one. If no candidate meets the criterion in a given batch, sampling continues iteratively until valid tail samples have been obtained for every position.

## B. Provable Security of MDDM

Assume that the distribution of carrier data is $P_C$, the distribution of encrypted data is $P_s$, and the relative entropy between carrier data and encrypted data is $D(P_c \| P_s)$, which is defined as:

$$D(P_C \| P_S) = \sum_{q \in \Omega} P_C(q) \lg \frac{P_C(q)}{P_S(q)} \tag{12}$$

If $D(P_C \| P_S) = 0$, the steganography system is said to be absolutely secure (Cachin, 1998; Weiming, 2023).

First, since the hidden message is not in the pixels of the image, it is impossible to find the hidden message through steganalysis in the pixels. Second, since the hidden message does not change the distribution of the noise, it will not cause any characteristic difference between the stego images and other naturally generated images. To sum up, MDDM is absolutely safe.

## C. Security of the Cardan Grille in MDDM

In MDDM, the security of the Cardan grille is closely related to the steganographic communication protocol. Specifically, if a new Cardan grille is used for each image, the distribution of the reversed noise for each image will be different, and third parties will not have sufficient information to infer whether an image contains hidden information. However, when multiple images share the same Cardan grille for steganography, the distribution patterns of their reversed noise will exhibit a certain degree of similarity. Therefore, we evaluated the privacy security of the Cardan grille steganographic method under the scenario where the Cardan grille is used for only one communication and then reused in subsequent transmissions, across different messages. The diffusion model employed is Stable Diffusion v2.1, with image resolution set to $512 \times 512$ and an empty prompt. We assume that an attacker has access to the stego images and is aware of the diffusion model, the steganographic method, and even the number of Cardan grille positions $l$ (i.e., the message length). The attacker's objective is to infer the position distribution and the ordering of the Cardan grille.

The attacker extracts the inverted noise from multiple images and filters out the elements with larger absolute values, as positions with larger absolute values are more likely to correspond to locations used by the Cardan grille. Two attack strategies are considered:

- Union Strategy: For each image, the attacker selects the $l$ position elements with the largest absolute values from the inverted noise vector. The positions chosen across all images are then combined via a union operation to obtain the final set of selected locations.

- Top Strategy: For each image, the attacker selects the $l$ position elements with the largest absolute values from the inverted noise vector. Subsequently, among all the selected positions from all images, the $l$ positions that occur most frequently are chosen as the final set of selected locations.

The experimental results are shown in Figure 8. "Union" refers to the position hit rate of the first strategy; "Union Redundancy" represents the redundancy of the union set from the first strategy (calculated as the ratio of the number of Cardan grille positions to the number of elements in the union set — lower values indicate higher redundancy); and "Top"

refers to the position hit rate of the second strategy. A high position hit rate in the Union strategy combined with low Union Redundancy indicates that the attacker has inferred a significant portion of the Cardan grille's position distribution.

The experimental results show that when the Cardan grille is used only once or reused up to twice, neither strategy poses a significant threat to the position distribution or ordering of the Cardan grille. When the Cardan grille is reused 3 to 5 times, the Union strategy achieves a higher hit rate, but the redundancy of the position set rapidly decreases, limiting the threat it poses to the Cardan grille's position distribution. The Top strategy also sees a modest improvement in hit rate, but neither strategy is able to compromise the ordering of the Cardan grille positions. When the Cardan grille is reused more than 5 times, the hit rate of the Union strategy shows no substantial improvement, while the position set redundancy continues to decline, paradoxically reducing its threat to the Cardan grille's positional information. The Top strategy shows only limited further improvement in hit rate, and neither strategy is ever able to compromise the ordering information of the Cardan grille.

Since the Cardan grille is ordered, the brute-force computational complexity of guessing the order of an n-position Cardan grille is $O(l!)$, where

$$l! \approx \sqrt{2\pi l}\left(\frac{l}{e}\right)^l, \tag{13}$$

according to Stirling's approximation. Since the difficulty of brute-forcing the ordering of the Cardan grille increases exponentially with the number of positions, a more secure strategy is to use longer messages. We therefore recommend that, in steganographic scenarios where the Cardan grille is reused, each image should carry at least 50 bits of hidden information.

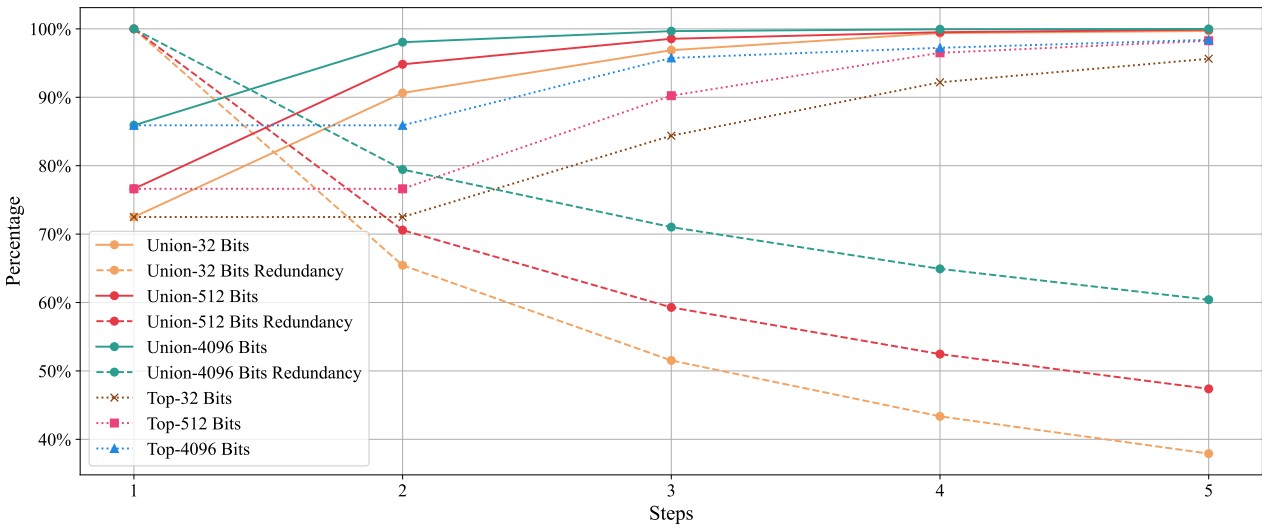

*Figure 8.* Simulated attacks against the MDDM that employs repeatedly used Cardan grille under varying message lengths based on Stable Diffusion v2.1. The percentages in the chart represent the proportion of simulated attack hit positions rather than the proportion of hit orders.

## D. Experiments

### D.1. Effects of Prompts on Inversion in Conditional Image Generation

In this experiment, we assess how prompts affect MDDM in conditional image generation, as shown in Figure 9. We use three evaluation metrics: (i) Case 1 quantifies the global discrepancy between the initial noise and the inverted noise using mean squared error (MSE); for comparability across metrics, we report $1 - \text{MSE}$. (ii) Case 2 computes, over all elements, the fraction of indices at which the initial and inverted noise have the same sign (either both positive or both negative), by element-wise comparison. (iii) Case 3 applies the same element-wise comparison but only at positions specified by the Cardan grille mask, which directly reflects MDDM performance. We further compare these three cases under two inversion

regimes: (a) inversion using null-text, and (b) inversion using the same prompt used during sampling. The results indicate that increasing the guidance scale drives the inverted noise further from the initial noise, while MDDM (Case 3) maintains high extraction accuracy. At the same time, inversion using null-text increases the discrepancy relative to the initial noise, yet MDDM remains consistently accurate. These findings suggest that, in conditional image generation, a receiver can reliably recover the hidden message solely via the Cardan grille without knowing the sender's prompt.

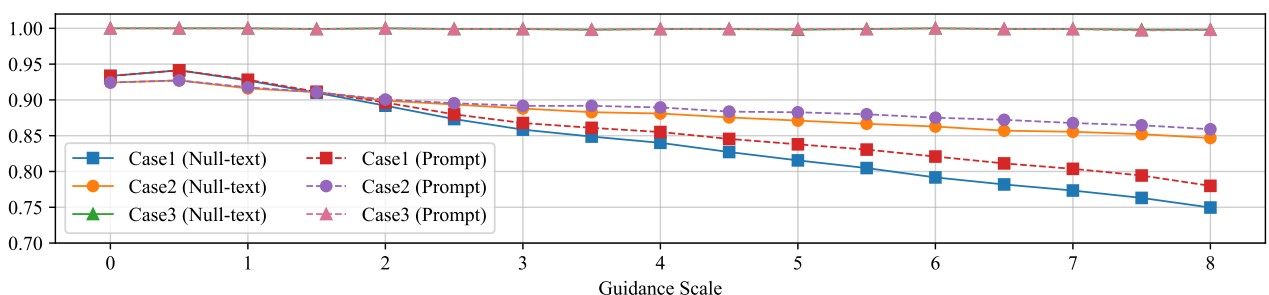

*Figure 9.* Effects of prompts on inversion in conditional image generation. Our method maintains high accuracy as the guidance scale increases.

### D.2. Quantitative Evaluation of Image Quality

We compare MDDM with two GAN-based generative image steganography methods, StegaStyleGAN (Su et al., 2024) and CIS-Net (You et al., 2022), on the CelebA dataset (Liu et al., 2015). The results are shown in Table 3. Due to image size limitations of CIS-Net, all comparisons are conducted on images of size 32 × 32. Moreover, considering the instability often associated with GAN training, and to avoid potential bias from our own reimplementation, we cite the results of StegaStyleGAN and CIS-Net directly from their original papers.

To quantitatively evaluate the image quality of our proposed method, we employ the Fréchet Inception Distance (FID), a widely used metric that assesses the similarity between the distributions of generated and real images. It is worth noting that FID scores are influenced by various factors, such as image compression and cropping.

Experimental results show that our method achieves competitive performance not only in terms of steganography capacity and accuracy but also in image quality, with FID scores comparable to those of existing methods. This indicates good generalization ability. Unlike GAN-based steganographic models, MDDM does not require additional training. Consequently, the image quality depends on the generative quality of the original diffusion model, and using more advanced diffusion models further improves the image quality produced by MDDM.

*Table 3.* Comparison of our method with baseline GAN methods.

| Method | Size | Capacity (Bpp)[*] | Accuracy | FID |
|---|---|---|---|---|
| CIS-Net (You et al., 2022) | 32 × 32 | 0.03 | 0.98 | 6.20 |
| StegaStyleGAN (Su et al., 2024) | 32 × 32 | 0.06 | 1.00 | 3.74 |
| MDDM-PU (Ours) | 32 × 32 | 0.08 | 0.99 | 11.71 |
| MDDM-PU3 (Ours) | 32 × 32 | 0.08 | 1.00 | - |

[*] Bpp means bits per pixel.

### D.3. Detailed Experimental Setup for Steganographic Capability

For unconditional image generation, we use pre-trained diffusion models on the CelebA-HQ (face images) (Karras et al., 2017), LSUN-Bedroom, and LSUN-Cat (Yu et al., 2015) datasets. For conditional image generation, we use Stable Diffusion v1.5. The prompts for the Face, Bedroom, and Cat categories are "Portrait photo, best quality, masterpiece, ultra detailed, UHD 4K, photographic, 1girl, face, looking at viewer, color photo, natural colors", "A photo of the bedroom", and "A cat", respectively.

As described in Pulsar (Jois et al., 2024), MDDM is also a symmetric key scheme. The sender can easily know the accuracy of the information extracted from the generated image and can terminate and regenerate. A key advantage of our method is that when the two parties communicate and determine the Cardan grille, the same secret message can be generated multiple times using different seeds without affecting the reception (see Appendix D.5 for details). Accordingly, we apply the MDDM-PU3 optimization strategy for unconditional pixel-space diffusion to mitigate potential instabilities (see Section 3.1) by selecting, from three consecutive samples, the image with the smallest information loss. This is completely feasible in practical applications, because if the improved algorithm in Appendix A is used, each unconditional generation and extraction for a $256 \times 256$ image takes only about 10 seconds on an NVIDIA RTX 3090.

### D.4. Quantitative Results Compared to Other Generative Image Steganography Methods

We conduct a quantitative comparison between our method and baseline methods on the FFHQ (Karras et al., 2019), LSUN-Bedroom, and LSUN-Cat (Yu et al., 2015) datasets, as shown in Table 4. Since neither StegaDDPM nor LDStega has made the experimental code or seed public, the data of StegaDDPM and LDStega are from the data in the paper LDStega (Peng et al., 2024) for reference. For Pulsar, we utilize its open-source code to conduct tests. The specific experimental settings of MDDM-PU3 are described in Appendix D.3. Using MDDM-PU3 can be regarded as comparing the accuracy after each seed sharing.

*Table 4.* Comparison of extraction accuracy and capacity between MDDM and baseline methods.

| Method | Type | Size | Capacity (Bits) | Accuracy | | |
|---|---|---|---|---|---|---|
| | | | | FFHQ | Bedroom | Cat |
| Pulsar (Jois et al., 2024) | | $256 \times 256$ | $\approx 4500$ | 0.9700 | 0.9300 | 0.9600 |
| StegaDDPM (Peng et al., 2023) | | $256 \times 256$ | 4096 | 0.9345 | 0.9019 | 0.9081 |
| LDStega (Peng et al., 2024) | GIS | $256 \times 256$ | 4096 | 0.9865 | 0.9850 | 0.9848 |
| MDDM-PU (Ours) | (Diffusion) | $256 \times 256$ | 4096 | 0.9248 | 0.9379 | 0.9219 |
| MDDM-PU3 (Ours) | | $256 \times 256$ | 4096 | 0.9865 | 0.9739 | **0.9989** |
| MDDM-LU (Ours) | | $256 \times 256$ | 4096 | **0.9950** | **0.9868** | - |

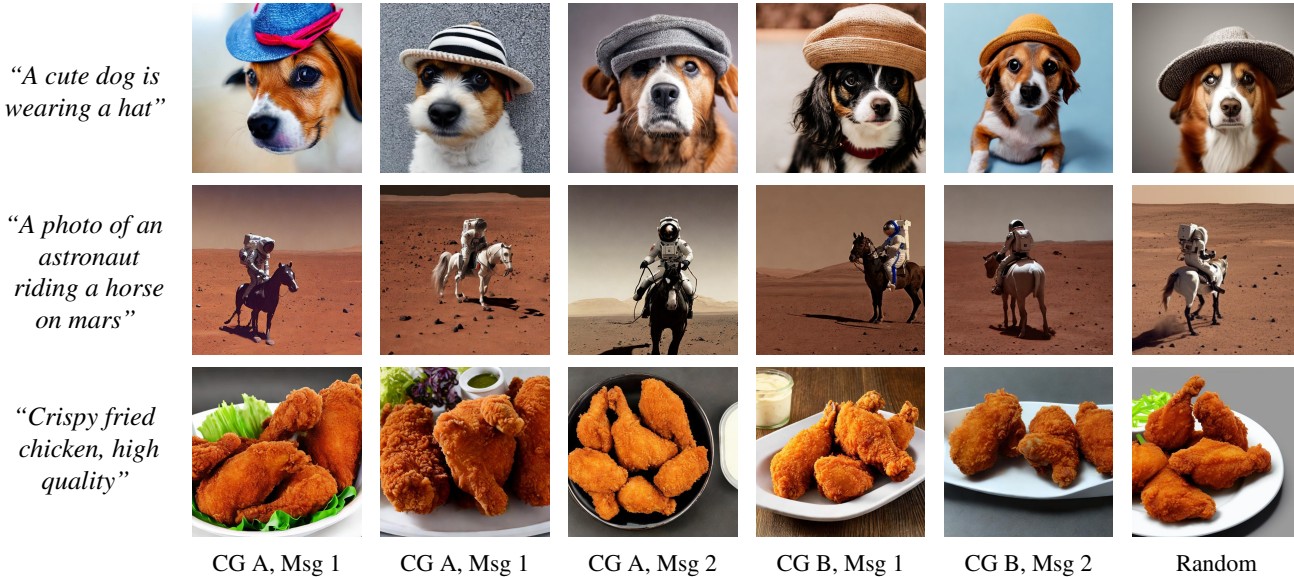

| | | | | | |
|---|---|---|---|---|---|
| CG A, Msg 1 | CG A, Msg 1 | CG A, Msg 2 | CG B, Msg 1 | CG B, Msg 2 | Random |

*"A cute dog is wearing a hat"*

*"A photo of an astronaut riding a horse on mars"*

*"Crispy fried chicken, high quality"*

*Figure 10.* Examples of images generated by Stable Diffusion v2.1 for various combinations of Cardan grilles, messages, and prompts (**CG** refers to the Cardan grille and **Msg** refers to the message). All images are $512 \times 512$ pixels with a hidden-message length of 1024 bits. Message-extraction accuracy for every case is 100%.

### D.5. Diversity of MDDM

Table 5 shows the comparison of our method with other methods in terms of diversity, where diversity refers to the number of images that can be generated after a single communication of a shared key. Figure 10 shows a visualization of the generation effect of MDDM in various cases.

*Table 5.* Comparison of the diversity of MDDM and baseline methods.

| Method | Hiding Type | Diffusion Type | Diversity (Number of Images) |
| --- | --- | --- | --- |
| StegaDDPM (Peng et al., 2023) | Binary | Pixel-Space | 1 |
| Pulsar (Jois et al., 2024) | Binary | Pixel-Space | 1 |
| LDStega (Peng et al., 2024) | Binary | Latent | 1 |
| CRoSS (Yu et al., 2024) | Image | Latent | Several[*] |
| MDDM (Ours) | Binary | Pixel-Space + Latent | $\infty$ |

[*] CRoSS can alter individual elements in the secret image but has difficulty changing the overall contents of the image.

