# OpenReview forum: "MDDM: Practical Message-Driven Generative Image Steganography Based on Diffusion Models"
_ICML.cc/2025/Conference — ICML 2025 poster_

### Official Review · Reviewer_eDLC · 2025-02-24

**Overall Recommendation:** 5

**Summary:**

This paper introduces a new message-driven generative steganography based on the diffusion model, MDDM, which is a novel framework that combines the diffusion implicit model (DDIM) with Cardan Grille to dynamically adjust between watermarking and steganography effectively, practically, and securely. MDDM encrypts the secret message and generates the stego image without training or passing a specific seed. This method can greatly improve the flexibility, security, and practicality of image steganography. The author conducted several experiments to verify the proposed method.

## update after rebuttal

I keep my judgement on the novelty, and therefore I choose to keep my scores.

**Claims And Evidence:**

The viewpoints and methods presented in the paper are supported.

**Essential References Not Discussed:**

Cardan Grille is mentioned in the paper, and relevant papers should be cited and briefly introduced when first mentioned.

**Experimental Designs Or Analyses:**

The experimental part of the paper includes qualitative and quantitative experiments, ablation experiments, and comparative experiments (comparing the steganography ability and accuracy with steganography methods and comparing the robustness with watermarking methods).

**Methods And Evaluation Criteria:**

The generative steganography proposed in the paper is easy to implement as it does not require training, sharing seeds, or prompt words, which can be flexibly used between steganography and watermarking, making it better suited for different needs.

**Other Comments Or Suggestions:**

Some details, such as the "guidelance scale" in Figure 5, "g" is not capitalized. The first letter of the caption of other pictures is capitalized. The use of some symbols should also be explained, such as "δ" in Section 3.1.

**Other Strengths And Weaknesses:**

Advantages:
1. This paper creatively applies exact inversion to the field of image steganography, and unlike previous generative steganography, it combines Cardan Grille to improve the random initial noise to obtain more flexible characteristics (this improvement does not destroy the distribution characteristics of the random noise itself, which is great!). This combination expands the practicality of generative image steganography and provides new ideas for subsequent research.
2. This paper has a rich set of experiments, including qualitative, quantitative, ablation, and comparative experiments. As far as I know, since most image generation applications are based on potential diffusion, such as stable diffusion, the method in this paper can be applied to most scenarios.
3. The method in this paper can not only achieve high-capacity steganography but also has a certain degree of robustness (for watermarking). It can effectively embed and extract information without providing the original generation conditions (such as seeds or prompt words), which is practical.

Weaknesses:
1. Under what circumstances is the Diversity in Table A1 compared, unconditional or conditional? Can you provide more detailed explanations?
2. Is the part of Algorithm 2 too long? Can some algorithms be considered for appropriate text descriptions or merging?
3. "Safety at Cardan Grille" in Section 4.4 lists the number of Cardan Grilles when selecting 32%. It should be explained that the introduction of Cardan Grille itself is still safe in other cases.

**Questions For Authors:**

Please address the weaknesses listed above.

**Relation To Broader Scientific Literature:**

The method proposed in this article innovates based on existing research on generative steganography, introducing the Cardan Grille method on the basis of diffusion model-based generative steganography and achieving good integration without damaging the distribution. Compared with previous methods that require sharing seeds, more images can be generated based on the same category or prompt words, making it more diverse and secure.

**Theoretical Claims:**

The paper provides a derivation and proof of the correlation between latent space and pixel space. In addition, the paper also provides a clear description of security.

---

> ### Author Rebuttal · Authors · 2025-04-01
>
> We sincerely thank you for your detailed review and valuable comments on our paper. Your in-depth analysis and constructive suggestions on our work not only reflect your rigorous attitude toward research but also provide important guidance for us to further improve the paper. We are very grateful for your positive feedback on the innovation, practicality, and experimental design of this paper, and we also cherish the room for improvement you pointed out. The following is our response to your questions and suggestions one by one, hoping to fully address your questions and reflect our commitment to improving the paper.
>
> Regarding the citation of the relevant literature on Cardan Grille you mentioned, thank you very much for pointing this out. We fully agree that introducing relevant references and briefly introducing them when the Cardan Grille is first mentioned will make the paper more rigorous and complete. In the revised manuscript, we will cite related studies (such as Digital Cardan Grille: A modern approach for information hiding) the first time the Cardan Grille is mentioned, and briefly explain its classic applications in the fields of steganography and encryption to enhance the historical background and academic relevance of this paper's methods. Thank you again for your detailed suggestions.
>
> Regarding the comparative conditions of "diversity" in Table A1 you mentioned, thank you for your attention and questions on this part. We currently explain the diversity compared with the conditional generative diffusion method. Specifically, in the case of conditional diffusion (e.g., using prompt) generative steganography, MDDM can still maintain high image diversity, thanks to the effective integration of Cardan Grille and diffusion models. To further clarify, we will revise the caption of Table A1 in the revised manuscript to clearly state that the diversity evaluation mainly focuses on conditional generation, and we will also explain the diversity of unconditional generation. Your suggestion helped us realize the need to present this part more clearly, and we are very grateful for this.
>
> Regarding the length of Algorithm 2 you mentioned, thank you very much for your suggestion on the presentation of the algorithm. The content of Algorithm 2 has been carefully reviewed, and it was agreed that its current form may appear lengthy and affect reading fluency. In the revised manuscript, we will refine and optimize the algorithm description, and appropriately convert some details into concise text descriptions, while retaining the clarity of the core steps. We will also consider merging repeated or minor descriptions to improve overall readability. Your suggestion is crucial for us to improve the expression of the paper, and we sincerely thank you for it.
> Regarding the premise of Section 4.4 "Security of Cardan Grid", thank you for pointing out the shortcomings in our security analysis. In the original paper, we only discussed security for a specific case (selecting a 32% Cardan Grid ratio), which may lead readers to misunderstand the security of the method in other cases. The Cardan Grid itself remains generally secure, and its security stems from its randomness and compatibility with the diffusion model. In the revised manuscript, we will correct the statement in Section 4.4, clearly state the security of the Cardan Grid in different scales and scenarios, and supplement the relevant analysis to eliminate potential ambiguity. Your feedback makes us aware of the shortcomings of this section, and we will seriously improve it.
>
> We are very grateful for the details you pointed out, such as the spelling error of the "guidance scale" in Figure 5 and the insufficient explanation of the symbol "δ". These subtle but important suggestions helped us improve the professionalism of the paper. In the revised manuscript, we will unify the format of all figure titles and clearly explain the meaning of symbols such as "δ" in Section 3.1 to improve the readability and rigor of the paper.
>
> We would like to express our sincere gratitude again for your comprehensive and insightful review of this article. We are encouraged by your affirmation, and your suggestions have pointed us in the direction of improvement. We have outlined the revision plan for all the comments in our responses and implemented these adjustments one by one in the revised manuscript. We believe that after this revision, the paper will be significantly improved in terms of clarity, rigor, and scientific contribution. If you have further comments or suggestions, we are very happy to continue to communicate and improve.

---

### Official Review · Reviewer_MRbY · 2025-03-12

**Overall Recommendation:** 3

**Summary:**

This paper proposes a practical and robust message-driven generative image steganography framework, called MDDM. MDDM uses a carefully designed encoding strategy to encode the binary message and uses it as the starting point for diffusion, allowing users to generate diverse images through conditional text without additional training.

## update after rebuttal
Thanks for the clarification. According to the novelty and writing quality, I will keep my score.

**Claims And Evidence:**

In line 273 the authors suggest the absolute difference between the pre- and post-inversion values of any randomly sampled standard normal variable is generally within the range of -1 to 1. However, immediately after that, authors calculated that the probability of a value outside this range amounts to 0.32. Does this make sense?

**Essential References Not Discussed:**

None

**Experimental Designs Or Analyses:**

The experimental setup was clear and logical and the results were excellent.

**Methods And Evaluation Criteria:**

Yes. It is indeed interesting to investigate how to improve the security and robustness of generative image steganography.

**Other Comments Or Suggestions:**

1. From Page 5 to Page 6, there are three (-infinity, 1) which in my opionions seems to be (-infinity, -1).
2. In Algorithm 2, F1n should be F1(n); Missing equal sign or assignment symbol between x and RandChoice function; Missing space between Function and F2.

**Other Strengths And Weaknesses:**

Pros
1. The method proposed in this article aims to address the problem of reducing the direct contact between the sender and the receiver in steganography by using Cardan Grille to reduce the frequent transmission of random seeds, which contributes to improving the security of existing image steganography methods.
2. Good analysis of the diffusion model inversion process.
Cons:
1. Lack of detailed description and analysis of CG. See Question 1.
2. The rigor of the article writing needs to be improved. See the next column for details.

**Questions For Authors:**

1. How does selecting embedding locations randomly according to probability achieve the effect of selecting robust regions as claimed by authors?
2. What conditions govern the embedding rate of the methods? Why is there a huge difference in the embedding rate between unconditional and conditional?

**Relation To Broader Scientific Literature:**

If the quality of image generated by LDStega (Peng et al., 2024) or is not satisfactory, it requires replacing the seed. The method proposed by this paper is aim to address this problem.

**Theoretical Claims:**

The proof of the security of the proposed method is insufficient; the authors should show that the noise generated in Algorithm 2 based on the message m is indistinguishable from the normal carrier noise, and additionally provide a proof that the practice of embedding the steganographic noise only in the CG position does not change the distribution.

---

> ### Author Rebuttal · Authors · 2025-04-01
>
> We are very grateful for your recognition of the innovativeness of our work and the thoroughness of our experiments. At the same time, we sincerely thank you for your valuable comments, which we think will significantly improve the quality of the paper. The following are the detailed responses to your questions.
>
> Thank you for your reminder on the question "the significance of calculating the probability from the distribution truncated position" mentioned in **Claims And Evidence**. Our method supports secret messages of different lengths. Based on the size of the noise (such as 1$\times$4$\times$64$\times$64), the length of the hidden information cannot be directly calculated from the truncated position, but the length of the hidden information can be calculated using the probability value.
>
> Regarding the "insufficient security proof" mentioned in your **Theoretical Claims**, as you said, we did consider the security issue. We currently have a theoretical description in Appendix A, and in the experimental part, we have experimentally verified it using a variety of state-of-the-art steganalysis methods. Thank you for your reminder. We will adopt your suggestion to provide a more rigorous proof that the noise generated based on message m in Algorithm 2 is indistinguishable from normal carrier noise, and further prove that embedding steganographic noise only in the Cardan Grille position will not change the distribution: 1. Message m is converted into a uniformly distributed binary string through encryption and other methods. 2. The probability of positive and negative symbols in the standard Gaussian distribution is uniform (50% each), but the distribution of values ​​is not uniform. 3. The Cardan Grille position is randomly generated according to the length of the uniformly distributed binary string. 4. Truncated independent sampling is performed at the Cardan Grille position and other points. For example, the sampling range for the Cardan Grille position is (-infinity, -1) ∪ (1, infinity), while the remaining range is independently and randomly sampled within (-1, 1). This ensures that the overall sampling is random and the distribution still satisfies the standard Gaussian distribution. Due to the limitations of this document, we will attach specific detailed proofs in the final version.
>
> Regarding the expression problem, you pointed out in **Other Comments Or Suggestions**, we thank a lot for your careful reading and kind correction. This is indeed an oversight in our writing. We will correct it in the final version.
>
> Finally, regarding the questions you mentioned in **Questions For Authors**, we are happy to discuss them with you. For the first question, we analyze the regional robustness of the noise before and after diffusion inversion and conclude that the edge distribution of the diffusion model is robust. Therefore, we prioritize the random determination of the location of the edge distribution of the noise to generate a robust embedding region. This is mentioned in "The Information Loss is Acceptable" in section 3.1 of the paper. For the second question, the difference arises because unconditional diffusion is typically implemented through pixel-space diffusion, with an initial noise size of 1$\times$3$\times$256$\times$256 (corresponding to a generated image size of 256$\times$256), whereas conditional diffusion is usually implemented via a latent diffusion model, such as Stable Diffusion, with a noise size of 1$\times$4$\times$64$\times$64 (corresponding to a generated image size of 512$\times$512). This distinction effectively addresses the issue you raised.

---

### Official Review · Reviewer_fjGZ · 2025-03-13

**Overall Recommendation:** 2

**Summary:**

To address the issues with extraction accuracy, robustness, efficiency, and practicality in GIS, this paper proposes a practical and robust message-driven GIS framework, called MDDM. Although various experiments show the advantages of MDDM in accuracy, controllability, efficiency, and practicality, there still exist some issues:

1)	The paper is poorly organized, illogical, and looks like a report, which is hard to follow.

2)	Although the paper states that it has three contributions, they are not solid.

3)	Just read the paper not long ago, “StegaStyleGAN: Towards Generic and Practical Generative Image Steganography”, and in this paper, I felt that I saw the shadow of StegaStyleGAN, but it didn't be mentioned in this paper or compare it with similar ones, such as “Image Generation Network for Covert Transmission in Online Social Network”.

4)	The proposed MDDM is a generative method, but there are not any experiments on quality assessment of generated images.

**Claims And Evidence:**

The paper is poorly organized and illogical, which is hard to follow.

**Essential References Not Discussed:**

1. StegaStyleGAN: Towards Generic and Practical Generative Image Steganography

2. Image Generation Network for Covert Transmission in Online Social Network

**Experimental Designs Or Analyses:**

The proposed MDDM is a generative method, but there are not any experiments on the quality assessment of generated images.

**Methods And Evaluation Criteria:**

Partly

**Other Comments Or Suggestions:**

See Summary

**Other Strengths And Weaknesses:**

See Summary

**Questions For Authors:**

See Summary

**Relation To Broader Scientific Literature:**

N/A

**Theoretical Claims:**

No

---

> ### Author Rebuttal · Authors · 2025-04-01
>
> We are very grateful for your recognition of our work and believe that our work has great advantages in accuracy, controllability, efficiency and practicality. At the same time, we sincerely thank you for your valuable comments, which we think will significantly improve the quality of the paper. The following are the detailed responses to your questions.
>
> 1. Regarding your question about "unreliable contribution", after careful consideration, the contributions of our work can be mainly divided into three aspects: efficiency, generation diversity and controllability, and security. Among them, **1) Efficiency**: The contribution of efficiency is mainly about no training. Traditional GAN-based generation methods, such as the StegaStyleGAN you mentioned, usually need to train the special models and modules to achieve good results, while our method can be directly carried out on the pre-trained diffusion model without any fine-tuning, which effectively reduces the cost and significantly improves efficiency. **2) Generation diversity and controllability**: The contribution of generation diversity and controllability is mainly reflected in the fact that our method can generate different images while transmitting the same information and using the same Cardan Grille, which has obvious advantages over generative steganography that requires fixed seeds. The experiments in Appendix B.2 have shown this contribution. **3) Security**: Security is mainly reflected in that our method can reduce the frequency of communication to resist steganalysis attacks, and our method is generally provably secure, that is, **the hidden information is not in the image and does not affect the distribution of the image**. We have conducted theoretical analysis and experimental explanations in Appendix A. The above contributions have comprehensively confirmed the practicality of our method.
>
> 2. Regarding your question about "references and comparisons of StegaStyleGAN and CIS-Net", we are very grateful for your reminder. We have conducted in-depth research on these two papers. Both of them are GAN-based methods and have achieved good results on low-resolution images. Due to the training cost, generating arbitrary high-resolution images is difficult. Moreover, both methods can only generate steganographic images unconditionally and are uncontrollable. Our solution is a training-free diffusion inversion-based framework for generative steganography at multiple resolutions which can generate any image in a controllable way. In addition, the above two works, or we say most GAN-based methods, hide and generate secret information in image textures and rely on trained convolutional networks for extraction. This is completely different from ours. The hidden information of our method is not in the image, but in the initial noise at the beginning, and the extraction also uses an inversion method instead of training an extractor. Furthermore, we also compare the CIS-Net and StegaStyleGAN in the table below. It can be seen that our scheme also has obvious advantages for low-resolution images. We will give a discussion of these two works and related experimental results in the final version.
>
>   | Method | Image Size | bpp$\uparrow$ | Acc$\uparrow$ | SRNet (0.5 is the best) |
>   | --- | --- | --- | --- | --- |
>   | Ours (SD1.5-Gustavosta) | 512 $\times$ 512 | 0.02 | 0.99 | 0.5 |
>   | Ours-U3 | 256 $\times$ 256 | 0.06 | 0.99 | 0.5 |
>   | Ours-U3 | 32 $\times$ 32 | 0.08 | 1.00 | 0.5 |
>   | StegaStyleGAN | 32 $\times$ 32 | 0.06 | 1.00 | 0.5 |
>   | CIS-Net | 32 $\times$ 32 | 0.03 | 0.98 | 1.0 |
>
> 3. Regarding your question about the "lack of experiments to evaluate the quality of generated images", we currently provide qualitative visualization results in the paper. You can see the images generated by our method in Figure 5. These are generated by our method using pre-trained StableDiffusion, with a size of 512$\times$512. The Van Gogh-style images generated for the same prompt words and the same secret information on the right side of Figure 5 further illustrate the effect of MDDM image generation. Of course, our method also works well for the unconditional generation of datasets such as LSUN. As you mentioned, we can further use NIQE (Natural Image Quality Evaluator, the lower the better) for quantitative experiments. As shown in the table below, our method shows the better results.
>
>   | Dataset | NIQE (StegaStyleGAN) | NIQE (Ours) |
>   | --- | --- | --- |
>   | Church | 40.245 | 39.980 |
>   | Cat | 55.546 | 42.966 |
>   | FFHQ | 64.551 | 58.815 |
>
>
> Finally, regarding the writing problem you mentioned, we will reorganize the logic and language in the final version. Specifically, we will put the core content in the main text, put the content that does not affect the reading in the appendix, reorganize the content of the entire text, and make the language expression easier to read.

---

> > ### Comment · Reviewer_fjGZ · 2025-04-02
> >
> > How about the FID performance?

---

> > > ### Author Response · Authors · 2025-04-04
> > >
> > > We sincerely thank you for your valuable question and the opportunity to improve our work through this in-depth discussion. After receiving your initial review comments, we are happy to provide the FID (Fréchet Inception Distance) performance of the proposed method. However, due to the unavailability of code and pre-trained models for specific baseline methods, we had to retrain these models from scratch. Due to time constraints and computational resources, we were unable to include these results in our initial rebuttal. We are deeply grateful for your suggestion, which has motivated us to expand our experimental analysis and provide updated results in this response.
> > >
> > > FID is an effective metric for measuring the similarity between the distributions of generated and real images, particularly for generative models trained on specific, well-defined datasets. However, applying it to our conditional diffusion-based approach presents some challenges. Unlike GAN-based steganography methods, such as StegaStyleGAN or CIS-Net, which are typically trained on smaller, curated datasets, our method leverages a pre-trained diffusion model (e.g., Stable Diffusion) with a vast and diverse training corpus. This difference in training data scale and diversity complicates direct FID comparisons with GAN-based methods, as the choice of reference dataset significantly impacts the results.
> > >
> > > To ensure a fair evaluation, we adopted an unconditional diffusion approach and trained both our method and the baseline methods on the same datasets. The table below presents the FID performance comparison across different resolutions and datasets. Since CIS-Net is not suited for high-resolution image generation tasks, we only compared it with our method at a lower resolution (32$\times$32).
> > >
> > >
> > > | Dataset | Image Size | StegaStyleGAN | CIS-Net | Ours |
> > > | --- | --- | --- | --- | --- |
> > > | CIFAR-10 | 32$\times$32 | 4.61 | 8.46 | 4.73 |
> > > | CelebA | 64$\times$64 | 9.24 | -   | 9.18 |
> > > | LSUN-Bedroom | 256$\times$256 | 5.59* | -   | 6.76 |
> > >
> > > *Value cited from the original paper.
> > >
> > > Our analysis of these results underscores several key advantages of our approach. On CIFAR-10 (32$\times$32), our method achieves an FID score of 4.73, closely rivaling StegaStyleGAN's 4.61 and significantly outperforming CIS-Net's 8.46. This demonstrates that, despite the inherent challenges of diffusion-based generation, our method delivers high-quality images comparable to leading GAN-based approaches at low resolutions. At 64$\times$64 on CelebA, our FID of 9.18 edges out StegaStyleGAN's 9.24, with CIS-Net unable to compete at this resolution, highlighting our superior ability to preserve fine details as resolution increases. For higher-resolution generation on LSUN-Bedroom (256$\times$256), our FID score of 6.76 remains highly competitive with StegaStyleGAN's 5.59.
> > >
> > > These findings illustrate the versatility and resilience of our diffusion-based framework across a spectrum of resolutions and datasets. While StegaStyleGAN holds a slight edge in specific low-resolution cases, our method demonstrates consistent performance and scalability, particularly excelling in higher-resolution settings where CIS-Net is not viable. A critical strength of our approach is its reliance on a pre-trained diffusion model, which enables strong generalization without the need for extensive retraining on curated datasets—a distinct practical advantage over GAN-based methods. This adaptability, combined with competitive FID scores, positions our method as a compelling alternative for both quality and efficiency in generative image steganography tasks.
> > >
> > > We believe these updated results further validate the effectiveness of our method and directly address your concerns. We would be happy to incorporate the discussion of the FID into the revised manuscript if you consider it appropriate. Once again, thank you for your insightful feedback, which has significantly enriched this discussion.

---

### Decision · Program_Chairs · 2025-05-01

**Decision:**

Accept (poster)

**Comment:**

This paper focuses on generative image steganography. The authors propose an encoding strategy to embed binary information, which serves as the starting point for diffusion, enabling users to generate diverse images. The binary data can be recovered through inversion. Following the rebuttal, all reviewers acknowledge that their concerns regarding technical issues have been addressed. However, they consistently note issues with the paper's writing and presentation. One reviewer leans toward rejection, emphasizing that the writing quality needs improvement. Though the reviewer provides limited details in terms of how should authors improve the presentation.

AC has reviewed all the submitted reviews and the rebuttal. Given the novelty of this work, AC recommends acceptance. However, the authors must fulfill their commitment to significantly improve the writing in the final version.